# TWO REGIMES OF GENERALIZATION FOR NON-LINEAR METRIC LEARNING

## ABSTRACT

A common approach to metric learning is to seek an embedding of the input data that behaves well with respect to the labels. While generalization bounds for linear embeddings are known, the non-linear case is not well understood. In this work we fill this gap by providing uniform generalization guarantees for the case where the metric is induced by a neural network type embedding of the data. Specifically, we discover and analyze two regimes of behavior of the networks, which are roughly related to the sparsity of the last layer. The bounds corresponding to the first regime are based on the spectral and $(2, 1)$-norms of the weight matrices, while the second regime bounds use the $(2, \infty)$-norm at the last layer, and are significantly stronger when the last layer is dense. In addition, we empirically evaluate the behaviour of the bounds for networks trained with SGD on the MNIST and 20newsgroups datasets. In particular, we demonstrate that both regimes occur naturally in realistic data.

## 1 INTRODUCTION

Metric Learning, Bellet et al. (2015), is the problem of finding a metric $\rho$ on the space of features, such that $\rho$ reflects some semantic properties of a given task. Generally, the input can be thought of as a set of labeled pairs $\{((x_i, x_i'), y_i)\}_{i=1}^n$, where $x_i, x_i' \in \mathbb{R}^d$ are the features, and $y_i$ is the label, indicating whether $x_i$ and $x_i'$ should be close in the metric or far apart. For instance, in face identification, Schroff et al. (2015), features $x_i$ and $x_i'$ corresponding to the same face should be close in $\rho$, while different faces should be far apart.

Note that the above metric learning formulation is fairly general and one can convert supervised clustering, or even standard classification problems into metric learning simply by setting $y_i = 1$ if $x_i$ and $x_i'$ have the same original label and $y_i = 0$ otherwise Davis et al. (2007); Weinberger & Saul (2009); Cao et al. (2016); Khosla et al. (2020); Chicco (2020).

The metric $\rho$ is typically assumed to be the Euclidean metric taken after a linear or non-linear *embedding* of the features. That is, we consider a parametric family $\mathcal{G}$ of embeddings into a $k$-dimensional space, $g : \mathbb{R}^d \to \mathbb{R}^k$, and set [1]

$$\rho(x, x') = \rho_g(x, x') = \|g(x) - g(x')\|_2^2. \tag{1}$$

As an example, Figure 1 shows tSNE plots, Maaten & Hinton (2008), of the classical 20newsgroups dataset. Figure 1a was generated using the regular bag of words representation of the data (see Sections 5 and D for additional details) while Figure 1b was generated by applying tSNE to an embedding of the data (of the form (3) below) learned by minimizing a loss based on labels, as described above. Clearly, there is no discernible relation between the label and the metric in the raw representation, but there is a strong relation in the learned metric. One can also obtain similar conclusions, and quantify them, by, for instance, replacing tSNE with spectral clustering.

The uniform *generalization* problem for metric learning is the following: Given a family of embeddings $\mathcal{G}$, provide bounds, that hold uniformly for all $g \in \mathcal{G}$, on the difference between the expected value of the loss and the average value of the loss on the train set; see Section 2 for a formal definition. Such bounds guarantee, for instance, that the train set would not be overfitted. Given a family

---

[1]Note that $\rho$ in (1) is not strictly a metric. Nevertheless, this terminology is common.

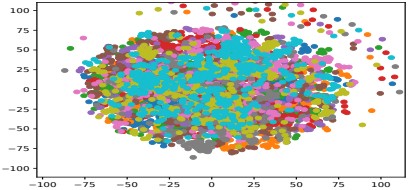 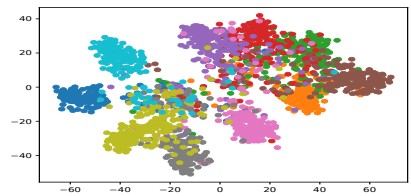

(a) Raw features, after normalization and 500 dim. PCA.

(b) Learned embedding, k=50, test set.

Figure 1: tSNE plot of the 20newsgroup data, restricted to the first 10 labels. Points are colored according to the label.

of embeddings $\mathcal{G}$, consider the family $\mathcal{F}$ of functions $f : \mathbb{R}^d \times \mathbb{R}^d \to \mathbb{R}$ of the form

$$\mathcal{F} = \{f(x, x') = \rho_g(x, x') \mid g \in \mathcal{G}\}. \tag{2}$$

We refer to these as the *distance functions*, which map a pair of features into their distance after the embedding. Well known arguments imply that one can obtain generalization bounds for $\mathcal{F}$ by providing upper bounds on the *Rademacher complexity* $\mathfrak{R}_n(\mathcal{F})$ of the family of the scalar functions $\mathcal{F}$. Therefore in what follows, we discuss directly the upper bounds on $\mathfrak{R}_n(\mathcal{F})$ for various families $\mathcal{G}$. We refer to Section 2 for formal definitions and details on the relation between $\mathcal{G}, \mathcal{F}, \mathfrak{R}_n(\mathcal{F})$, and generalization bounds.

## 1.1 OVERVIEW OF THE RESULTS

In this paper our goal is to study generalization bounds for neural network type non-linear embeddings $g : \mathbb{R}^d \to \mathbb{R}^k$ given by neural networks of depth $L \geq 1$. Specifically, we consider embeddings of the form

$$g(x) = g_A(x) = \phi_L\left(A_L^t \cdot \left(\phi_{L-1}\left(A_{L-1}^t \ldots \phi_1\left(A_1^t x\right) \ldots\right)\right)\right), \tag{3}$$

where $A = (A_1, \ldots, A_L)$ is a tuple of matrices such that $A_i \in \mathbb{R}^{k_{i-1} \times k_i}$ for $i = 1, \ldots, L$, and $k_i$ are the layer widths, with $k_0 = d$ and $k_L = k$. The activations $\phi_i : \mathbb{R} \to \mathbb{R}$ are assumed to be Lipschitz, with $\|\phi\|_{Lip} \leq \rho_i$, and act on vectors in $\mathbb{R}^{k_i}$ coordinatewise. A family of matrix tuples will be generally denoted by $\mathcal{A}$, while the associated embeddings family (3) will be denoted by $\mathcal{G}$ and the associated family of distance functions, (2), will be denoted by $\mathcal{F}$.

In the context of deep learning generalization guarantees for classification, there exists a large body of work on various complexity bounds related to the family of mappings $\mathcal{G}$. The current strongest uniform bounds were given in Bartlett et al. (2017) (see also Neyshabur et al. (2018)). Our goal here is to translate these bounds to bounds on the Rademacher complexity of the family $\mathcal{F}$, derived from $\mathcal{G}$. That is, we study the aspects of the problem that are specific to the metric learning setting.

To state the results, we require a few norm definitions: For matrix $A \in \mathbb{R}^{s \times t}$, $\|A\|_{op}$ is the spectral norm, and set $\|A\|_{2,1} = \sum_{i=1}^{t} \|A_{\cdot i}\|_2$, where $A_{\cdot i}$ is the $i$-th column of $A$. For a family $\mathcal{A}$ of matrix tuples, $A = (A_1, \ldots, A_L) \in \mathcal{A}$, for $i \leq L$ we set

$$\|\mathcal{A}_i\|_{op} = \sup_{A \in \mathcal{A}} \|A_i\|_{op} \text{ and } \|\mathcal{A}_i\|_{2,1} = \sup_{A \in \mathcal{A}} \|A_i\|_{2,1}. \tag{4}$$

Thus $\|\mathcal{A}_i\|_{op}, \|\mathcal{A}_i\|_{op}$ are the largest respective norms of the component $A_i$ in $\mathcal{A}$.

With this notation, our first result is the following bound (up to logarithmic factors):

$$\mathfrak{R}_n(\mathcal{F}) \leq \overline{O}\left(\frac{1}{\sqrt{n}} b^2 \left(\prod_{i=1}^{L} \rho_i \|\mathcal{A}_i\|_{op}\right)^2 \left(\sum_{i=1}^{L} \frac{\|\mathcal{A}_i\|_{2,1}^{\frac{2}{3}}}{\|\mathcal{A}_i\|_{op}^{\frac{2}{3}}}\right)^{\frac{3}{2}}\right). \tag{5}$$

Here $b$ is an upper bound on inputs, $\|x_i\|_2, \|x_i'\|_2 \leq b$ for $i \leq n$. The full statement is given in Section 4, Theorem 1. This bound can be regarded as a natural extension of the bounds of Bartlett

et al. (2017) to the metric learning setting. Indeed, it uses the same parameters – the $\|A\|_{op}$ and $\|A\|_{2,1}$ norms. The quantity $\left(\prod_{i=1}^{L}\rho_i\|\mathcal{A}_i\|_{op}\right)$ is the maximal Lipschitz constant of the mappings in $\mathcal{G}$, and it will typically be the dominant term in (5). One of the appealing properties of (5), inherited from the bounds in Bartlett et al. (2017), is that it is *dimension free*, in the sense that the depth $L$ and the the layer widths $k_i$ do not enter the bound explicitly.

As noted above, for the case of classification, the bounds in Bartlett et al. (2017) are currently the strongest known uniform bounds. However, we now show that in the metric learning setting, one can improve the bound in some situations, by using the $\|A\|_{2,\infty} = \max_{i\leq k}\|A_{.i}\|_2$ norm at the last layer. In Theorem 2, Section 4, we show the following:

$$\mathfrak{R}_n\left(\mathcal{F}\right) \leq \overline{O}\left(\frac{k}{\sqrt{n}}b^2\left(\rho_L\|\mathcal{A}_L\|_{2,\infty}\prod_{i=1}^{L-1}\rho_i\|\mathcal{A}_i\|_{op}\right)^2\left(\sum_{i=1}^{L-1}\frac{\|\mathcal{A}_i\|_{2,1}^{\frac{2}{3}}}{\|\mathcal{A}_i\|_{op}^{\frac{2}{3}}}+1\right)^{\frac{3}{2}}\right). \tag{6}$$

We refer to (5) as the *sparse* bound, and to (6) as the *non-sparse* bound. To compare the bounds, and to see the relation to sparsity, let us first consider the simpler single layer setting, $L=1$. In this case, the bounds read

$$\mathfrak{R}_n\left(\mathcal{F}\right) \leq O\left(\frac{b^2\|\phi\|_{Lip}^2\|\mathcal{A}\|_{op}\|\mathcal{A}\|_{2,1}}{\sqrt{n}}\right), \tag{7}$$

and

$$\mathfrak{R}_n\left(\mathcal{F}\right) \leq O\left(\frac{kb^2\|\phi\|_{Lip}^2\|\mathcal{A}\|_{2,\infty}^2}{\sqrt{n}}\right), \tag{8}$$

where $\mathcal{A}$ is the family of weights matrices, $A \in \mathbb{R}^{d\times k}$, $\phi$ an activation, and $\|\mathcal{A}\|_* = \sup_{A\in\mathcal{A}}\|A\|_*$ where $\|\cdot\|_*$ is one of $\|\cdot\|_{2,1}, \|\cdot\|_{2,\infty}, \|\cdot\|_{op}$. Consider first the case where $\mathcal{A}$ contains non sparse matrices $A \in \mathcal{A}$. These are the weights $A$ where all $k$ neurons have roughly the same norm, or equivalently,

$$\|A\|_{2,1} \sim k\|A\|_{2,\infty}. \tag{9}$$

We refer to this condition as the *dense regime*. Note that (9) holds true for neural networks at initialization, and in Section 5 we observe empirically that this also holds for SGD trained networks on MNIST data. Next, when (9) holds, we also have that

$$\|A\|_{op} \sim \sqrt{k}\|A\|_{2,\infty}. \tag{10}$$

Indeed, one has $\|A\|_{op} \leq \sqrt{k}\|A\|_{2,\infty}$ for any $A$, and (9) implies $\|A\|_{op} \geq const\cdot\sqrt{\frac{k}{d}}\|A\|_{2,\infty}$ (see Lemma 3 in supplementary material A). The condition (10) will be also experimentally verified for MNIST data. Now, substituting (9),(10) into (7) and (8), we obtain that in the non-sparse regime, (8) is stronger than (7) by a factor of $\sqrt{k}$, a significant improvement. On the other hand, when $A$ is sparse, with most output neurons zeroed out (e.x. $\|A\|_{2,1} = \|A\|_{op} = \|A\|_{2,\infty}$), (7) will be much stronger. Such sparse networks can be achieved by adding explicit sparsity regularization terms to the cost. This will be the case for networks trained on the 20newsgroups dataset.

Finally, as evident from (5) and (6), for the multilayer case $L > 1$, the relation between the sparse and non-sparse bounds is more involved. However, the asymptotics of the dependence on $k$ is similar: With all other $k_i$ fixed, (6) will be better by a factor of $\sqrt{k}$ for non-sparse last layer $A_L$, and (5) will be better otherwise.

We now discuss an additional direction in which the bounds may be improved: Note that the dependence of the bound on the coefficients of the weights in (5) and (6) is *quadratic*. As we show in Section 4, this is in general unavoidable, due to the quadratic form of the metric (1). However, if the highest activation, $\phi_L$ is *bounded*, such as for instance the sigmoid $\phi(x) = (1+e^{-x})^{-1}$, then one can obtain a linearly homogeneous dependence on the layer weights, thus potentially significantly improving the bounds. These bounds are given in (18) and (21) in Theorems 1 and 2 respectively. For comparison, in the single layer case the sparse and non-sparse bounded $\phi_L$ bounds are given by

$$\mathfrak{R}_n\left(\mathcal{F}\right) \leq O\left(\frac{\sqrt{k}b\|\phi\|_\infty\|\phi\|_{Lip}\|\mathcal{A}\|_{2,1}}{\sqrt{n}}\right) \text{ and } \mathfrak{R}_n\left(\mathcal{F}\right) \leq O\left(\frac{kb\|\phi\|_\infty\|\phi\|_{Lip}\|\mathcal{A}\|_{2,\infty}}{\sqrt{n}}\right),$$
$$\tag{11}$$

where $\|\phi\|_\infty$ is the upper bound on the values of $\phi$.

## 1.2 METHODS

The general proof strategy we take in this paper is to combine the covering number bounds for the family $\mathcal{G}$, due to Bartlett et al. (2017), Zhang (2002), with the analysis of the specific structure induced by the metric learning problem, i.e. the structure of the family $\mathcal{F}(\mathcal{G})$, when $\mathcal{G}$ is fixed.

Theorems 1 and 2 exploit this structure differently. In Theorem 1 we directly estimate the covering numbers of $\mathcal{F}$ in terms of those of $\mathcal{G}$. Note that, for instance, $\mathcal{F}$ and $\mathcal{G}$ are function families on different domains, and thus such estimates require some care. Once the estimates are obtained, however, we bound $\mathfrak{R}_n(\mathcal{F})$ using a standard Dudley entropy integral argument (Vershynin (2018)).

On the other hand, to prove Theorem 2, we use the special structure of the metric (1) as a *sum*. This allows a decomposition of the problem into $k$ problems, each of which has $k = 1$, and yields bounds which depend on $\|A\|_{2,\infty}$ and $k$, rather than on $\|A\|_{op}, \|A\|_{2,1}$. While this may seem as a rougher bound at first glance, as discussed above it is in fact much stronger than Theorem 1 in some situations.

Finally, we note that up to now generalization for metric learning was only studied in the linear case (see Section 3 for a detailed discussion). In particular, uniform bounds were derived in Verma & Branson (2015) and in Cao et al. (2016). Interestingly, already in the basic situation where $L = 1$ but $\phi(x) \neq x$ (i.e. a single layer but non-linear activation), one can not deduce our bounds (7) or (8), for instance, from the bounds in Verma & Branson (2015) or Cao et al. (2016). Nor one can use the methods employed in these works to obtain such bounds. The reason for this is that there seems to be no simple principle that would "remove the linearity" (such as the Contraction Lemma, Mohri et al. (2018) ) in the case of the cost (1). As a result, somewhat surprisingly, even to obtain the linear instead of quadratic homogenity, as in (11) for $L = 1$, it appears that one already requires the methods used in this paper.

To summarize, we have obtained the first uniform generalization guarantees for multilayer non-linear metric learning. In particular, we introduced two types of bounds, which are appropriate for sparse and non-sparse weights $A_L$. We have also shown that by using a bounded last layer activation, one may avoid the quadratic dependence of the bound on the parameters, and we have empirically verified that both sparse and dense regimes may occur in SGD optimized networks.

The rest of this paper is organized as follows: In Section 2 we overview the necessary background on metric learning and generalization. Literature and related work are discussed in Section 3. In Section 4 we give the full formal statements of the results, and overview the main ideas involved in the proofs. Full proofs are deferred to the supplementary material due to space constraints. Experiments are described in Section 5 and concluding remarks are given in Section 6.

## 2 METRIC LEARNING BACKGROUND

In this paper we are assuming that the training data is given as a set $\{((x_i, x_i'), y_i)\}_{i=1}^n$ of labeled feature pairs, which we assume to be sampled independently form some distribution $\mathcal{D}$ on $\mathbb{R}^d \times \mathbb{R}^d \times Y$, where $Y$ is some set of label values. It is usually sufficeint to take $Y = \{0, 1\}$. Note that there may be dependence *within* the pair, i.e. $x_i'$ may depend on $x_i$.

The quality of the embedding $g$ on a data point $((x, x'), y)$ is measured via a *loss function* $\ell$, which usually depends on $((x, x'), y)$ only through the metric. That is, given an embedding $g \in \mathcal{G}$, let $f \in \mathcal{F}$ be the corresponding distance function, $f(x, x') = \rho_g(x, x')$ (see (2)). We assume that there is a fixed function $\ell : \mathbb{R} \times Y \to [0, 1]$, such that the loss of the distance function $f \in \mathcal{F}$ on a data point $((x, x'), y)$ is given by $L_f((x, x'), y) = \ell(f(x_i, x_i'), y_i) = \ell(\rho_g(x_i, x_i'), y_i)$. A typical example of a loss $\ell$ is the following version of the margin loss:

$$\ell_{S,D}(a, y) = \begin{cases} ReLU(a - S) & \text{if } y = 1 \\ ReLU(D - a) & \text{otherwise,} \end{cases} \tag{12}$$

where $ReLU(x) = \max(0, x)$. As discussed in Section 1, this loss embodies the principle that $(x, x')$ should be close iff $y = 1$, by penalizing distances above $S$ when $y = 1$ and penalizing distances below $D$ otherwise.

The overall loss on the data, or the empirical risk, is given by

$$\hat{R}_f = \frac{1}{n} \sum_{i=1}^{n} L_f((x_i, x_i'), y_i). \tag{13}$$

The expected risk is given by $R_f = \mathbb{E}_{((x,x'),y) \sim \mathcal{D}} L_f((x, x'), y)$. The uniform generalization problem of metric learning is similar to the generalization problem of classification: One is interested in conditions on the family $\mathcal{F}$ under which the gap between expected and empirical risks, $R_f - \hat{R}_f$, is small for all $f \in \mathcal{F}$.

Finally, since $\{((x_i, x_i'), y_i)\}_{i=1}^{n}$ are independent, standard results imply that to control the uniform generalization bounds of the risk, it is sufficient to control the Rademacher complexity of the family $\mathcal{F}$. Specifically, we have that (see Mohri et al. (2018) Theorem 3.3 and Lemma 5.7),

$$\sup_{f \in \mathcal{F}} R_f - \hat{R}_f \leq 2 \|\ell\|_{Lip} \mathfrak{R}_n(\mathcal{F}) + \sqrt{\frac{\log \delta^{-1}}{2n}}$$

holds with probability at least $1 - \delta$. Here $\mathfrak{R}_n(\mathcal{F})$ is the Rademacher complexity of $\mathcal{F}$ and $\|\ell\|_{Lip}$ is the Lipschitz constant of $\ell$ as a function of its first coordinate. In particular, we have $\|\ell\|_{Lip} = 1$ for losses defined by (12).

## 3 LITERATURE

General surveys of the field of metric learning can be found in Bellet et al. (2015) and Kulis (2012). See also Chicco (2020) for a survey of recent applications in deep learning contexts. In these situations, the metric learning loss is sometimes referred to as a Siamese network.

Up to now, generalization guarantees in metric learning were only studied in the *linear* setting, i.e. for embeddings of the form (3) where $L = 1$ and $\phi(x) = x$. In particular, all literature cited in this section deals with the linear case.

As discussed in Sections 1 and 2, in this paper we use the formal metric learning setting introduced in Verma & Branson (2015), where we assume that the data comes as a set of iid feature pairs with a label per pair, $((x_i, x_i'), y_i)_{i=1}^{n}$. In this setting, the empirical risk is given by (13). In the special case where the data comes with a label per feature, $(x_i, l_i)_{i=1}^{n}$, one can use an alternative notion of empirical risk, given by

$$\tilde{R}_f = \frac{1}{n(n-1)} \sum_{i=1}^{n} \sum_{j \neq i} L_f\left((x_i, x_j), \mathbb{1}_{\{l_i \neq l_j\}}\right), \tag{14}$$

which is viewed as a second order U-statistic, see Cao et al. (2016). That is, instead of considering the input as a set of independently sampled pairs, which can be obtained from the $(x_i, l_i)_{i=1}^{n}$ by, for instance, creating pairs out of consecutive samples, in (14) one considers all possible pairs. On one hand, compared to the risk $\hat{R}_f$ defined by (13), for small datasets $\tilde{R}_f$ might make a somewhat better use of the data. On the other hand, $\tilde{R}_f$ is less general, since the data does not necessarily have to be generated by the label-per-feature setting. Indeed, consider the celebrated word2vec text embedding technique, Mikolov et al. (2013b), Mikolov et al. (2013a). In word2vec, tokens $x$ and $x'$ should be mapped to similar vectors if $x$ and $x'$ tend to appear as contexts of each other, and should be mapped to distant vectors otherwise. Equivalently, similar token pairs are extracted from "windows" of the text, while dissimilar pairs are sampled "contrastively", i.e. i.i.d from a (version of) the marginal distribution. Clearly one can model this using the general setting described in Section 2, adopted in this paper. However, note that there are no labels $l$ one can attach to the tokens $x$ such that the distribution $((x, x'), y)$ can be described via the label-per-feature as above, with i.i.d samples $(x, l)$. Moreover, even when the label-per-feature setting is applicable, evaluating (14) may be computationally difficult since the number of terms in the sum is quadratic in dataset size.

The generalization framework considered in this paper is that of the *uniform generalization bounds*, see Mohri et al. (2018). Alternatively, in Wang et al. (2019), Lei et al. (2020), the linear case of metric learning was studied in the framework of algorithmic stability (Bousquet & Elisseeff (2002), Feldman & Vondrak (2019), Bousquet et al. (2020)). In these works, stability, and consequently

generalization bounds, were obtained for an appropriately regularized empirical risk minimization (ERM) procedure. We note that these methods rely strongly on the (uniform) convexity of the underlying problem, and are unlikely to be generalized to non linear settings. In Bellet & Habrard (2015),Christmann & Zhou (2016), linear metric learning was studied in the framework of algorithmic robustness, Xu & Mannor (2012).

Finally, uniform bounds for the linear case were studied in Verma & Branson (2015), and similar results for the version of risk as in (14) were obtained in Cao et al. (2016). In particular it was shown in Verma & Branson (2015) that

$$\mathfrak{R}_n\left(\mathcal{F}\right) \leq \frac{b^2 \sup_{(A_1) \in \mathcal{A}} \left\| A_1 A_1^t \right\|_F}{\sqrt{n}}, \tag{15}$$

where $\left\| \cdot \right\|_F$ is the Frobenius norm. Interestingly, when $\phi(x) = x$, one can derive the single layer inequalities (7) and (8) from (15). Indeed, since $\left\| \cdot \right\|_F$ is a *matrix norm*, Bhatia (1997), we have $\left\| A A^t \right\|_F \leq \left\| A \right\|_{op} \left\| A^t \right\|_F = \left\| A \right\|_{op} \left\| A \right\|_F$ for any $A$. This in turn can be bounded either as $\left\| A \right\|_{op} \left\| A \right\|_F \leq \left\| A \right\|_{op} \left\| A \right\|_{2,1}$ or as $\left\| A \right\|_{op} \left\| A \right\|_F \leq \sqrt{k} \left\| A \right\|_{2,\infty} \cdot \sqrt{k} \left\| A \right\|_{2,\infty}$. [2] However, as discussed in Section 1.2, there likely is no way to obtain the general (i.e. non-linear) statements (7) and (8) directly from the purely linear (15).

The bound (15) itself is derived in Verma & Branson (2015) using a relatively short elegant argument involving only the Cauchy Schwartz inequality. However, again, this argument can only be applied when $\phi$ is the identity. Similarly to the situation in classification, for the non-linear settings other arguments are required.

## 4 RESULTS

In this Section we introduce the full statements of the main results, Theorems 1 and 2, and overview the main ideas of the proofs. The full proofs are given in supplementary material Sections B and C.

### 4.1 NOTATION

We begin with some necessary notation. For a vector $v = (v_1, \ldots, v_m) \in \mathbb{R}^m$, the $\ell_p$ norm is denoted by $\left\| v \right\|_p = \left( \sum_{j=1}^m |v_j|^p \right)^{1/p}$. For a matrix $A \in R^{d \times k}$, and $1 \leq p, s \leq \infty$, denote $\left\| A \right\|_{p,s} = \left\| \left( \left\| A_{\cdot 1} \right\|_p, \ldots, \left\| A_{\cdot k} \right\|_p \right) \right\|_s$. That is, one first computes the $p$-th norm of the columns and then the $s$-th norm of the vector of these norms. Note that $\left\| A \right\|_{2,2} = \left\| A \right\|_2$ is the Frobenius norm. Denote by $\left\| A \right\|_{op}$ the spectral norm of $A$. Throughout $\phi : \mathbb{R} \to \mathbb{R}$ will denote an activation/non-linearity, with Lipschitz constant $\left\| \phi \right\|_{Lip}$ and such that $\phi(0) = 0$.

The input features $\{(x_i, x_i')\}_{i=1}^n \subset \mathbb{R}^d \times \mathbb{R}^d$, (see Sections 1,2) may be alternatively organized as two matrices, $X, X' \in \mathbb{R}^{n \times d}$, with rows $x_i$ and $x_i'$ respectively. If $\mathcal{F}$ is a family of functions from $\mathbb{R}^d \times \mathbb{R}^d$ to $\mathbb{R}$, the Rademacher complexity of $\mathcal{F}$ *given the inputs* $X, X'$ is defined by

$$\mathfrak{R}_n\left(\mathcal{F}\right) = \mathfrak{R}_n\left(\mathcal{F}, X, X'\right) = \frac{1}{n} \mathbb{E}_\sigma \sup_{f \in \mathcal{F}} \sum_{i=1}^n \sigma_i f(x_i, x_i'), \tag{16}$$

where $\sigma_i$ are independent Bernoulli variables with $\mathbb{P}\left(\sigma_i = 1\right) = \mathbb{P}\left(\sigma_i = -1\right) = \frac{1}{2}$.

Finally, recall that if $\mathcal{A}$ is a family of matrix tuples, $A = (A_1, \ldots, A_L)$, and $\left\| \cdot \right\|_*$ some norm on matrices, then for $i \leq L$, $\left\| \mathcal{A}_i \right\|_*$ is defined as $\left\| \mathcal{A}_i \right\|_* = \sup_{A \in \mathcal{A}} \left\| A_i \right\|_*$.

### 4.2 STATEMENTS AND DISCUSSION

We have the following bound:

---

[2] $\left\| A \right\|_{op} \leq \sqrt{k} \left\| A \right\|_{2,\infty}$ is shown in supplementary material Section A

**Theorem 1.** *Let $\mathcal{A} = \{(A_1, \ldots, A_L)\}$ be a family of matrix tuples, $A_i \in \mathbb{R}^{k_{i-1} \times k_i}$, such that $k_0 = d$ and $k_L = k$. Denote $W = \max_{1 \le i \le L} k_i$. Let $\mathcal{G}$ be the associated family of embeddings (3), and $\mathcal{F}$ the associated family of distance functions, (2). Let $\{(x_i, x_i')\}_{i=1}^n$ be inputs, such that $\|x_i\|_2, \|x_i'\|_2 \le b$ for all $i \le n$. Set $\alpha_i = \|\mathcal{A}_i\|_{op}$, $\beta_i = \|\mathcal{A}_i\|_{2,1}$ and $\rho_i = \|\phi_i\|_{Lip}$ for $i \le L$. Then*

$$\mathfrak{R}_n(\mathcal{F}) \le O\left( \frac{1}{n} + \frac{\log(2W)\left[\log n + 2\log\left(4b\left(\prod_{i=1}^L \rho_i \alpha_i\right)\right)\right]}{\sqrt{n}} b^2 \left(\prod_{i=1}^L \rho_i \alpha_i\right)^2 \left(\sum_{i=1}^L \frac{\beta_i^{\frac{2}{3}}}{\alpha_i^{\frac{2}{3}}}\right)^{\frac{3}{2}} \right).$$

(17)

*Alternatively,*

$$\mathfrak{R}_n(\mathcal{F}) \le O\left( \frac{1}{n} + \frac{\sqrt{k}\|\phi_L\|_\infty b \log(2W)\left[\log n + 2\log(4\sqrt{k}\|\phi_L\|_\infty)\right]}{\sqrt{n}} \left(\prod_{i=1}^L \rho_i \alpha_i\right) \left(\sum_{i=1}^L \frac{\beta_i^{\frac{2}{3}}}{\alpha_i^{\frac{2}{3}}}\right)^{\frac{3}{2}} \right).$$

(18)

The general structure of this bound is inherited from the covering number bounds of $\mathcal{G}$ in Bartlett et al. (2017). In contrast to the bounds in classification, note that the dependence on the coefficients of $A_i$ and on $b$ is *quadratic*. This can not in general be improved upon. Indeed, consider the special case of RelU activations, $\phi_i(x) = \max\{0, x\}$. Then the family $\mathcal{G}$ is positively homogeneous with respect to the matrix weights: Let $u = (u_1, \ldots, u_L)$ be a tuple of scalars, $u_i \ge 0$, and if $A = (A_1, \ldots, A_L)$ denote by $uA$ the matrix tuple $(u_1 A_1, \ldots, u_L A_L)$. Recall that $g_A$ is the embedding induced by $A$, given by (3). Then we have

$$g_{uA} = \left(\prod_{i=1}^L u_i\right) g_A.$$

(19)

Since the definition of $\mathcal{F}$ involves the squared norm, it follows from (19) that $f_{uA} = \left(\prod_{i=1}^L u_i\right)^2 f_A$ and thus $\mathfrak{R}_n(\mathcal{F})$ must be quadratically homogeneous in each of $u_i$.

On the other hand, as discussed in Section 1.1, when $\phi_L$ is bounded, it is possible to avoid the quadratic dependence. In this case we have the inequality (18) in Theorem 1.

We now briefly sketch the proof of Theorem 1. The argument is based on bounding the covering numbers $\mathcal{N}(\mathcal{F}_{X,X'}, \varepsilon)$ of the set $\mathcal{F}_{X,X'} \subset \mathbb{R}^n$ – the restriction of $\mathcal{F}$ to the input. As mentioned in Section 1.2, given such bounds, we bound $\mathfrak{R}_n(\mathcal{F})$ by the Dudley entropy integral. To bound $\mathcal{N}(\mathcal{F}_{X,X'}, \varepsilon)$, we represent $\mathcal{F}_{X,X'}$ as (a subset of) a Lipschitz image of the Cartesian product $\mathcal{G}_X \times \mathcal{G}_{X'}$, i.e. $\mathcal{G}_X \times \mathcal{G}_{X'} \overset{\Psi}{\longmapsto} \mathcal{F}_{X,X'}$ for an appropriate mapping $\Psi$. Here $\mathcal{G}_X$ and $\mathcal{G}_{X'}$ are the restrictions of $\mathcal{G}$ to the inputs. The covering numbers of $\mathcal{G}_X$ where estimated in Bartlett et al. (2017), and using these estimates we derive bounds on the coverings of $\mathcal{G}_X \times \mathcal{G}_{X'}$ and consequently of $\mathcal{F}_{X,X'}$. The full details are given in Section B. Here we note that the Lipschitz constant of the mapping $\mathcal{G}_X \times \mathcal{G}_{X'} \mapsto \mathcal{F}_{X,X'}$ can be estimated in two different ways, depending on whether $\phi_L$ is bounded, which result in the bounds (17) and (18).

We now state our second main result.

**Theorem 2.** *Let $\mathcal{A} = \{(A_1, \ldots, A_L)\}$ be a family of matrix tuples, $A_i \in \mathbb{R}^{k_{i-1} \times k_i}$, such that $k_0 = d$ and $k_L = k$. Denote $W = \max_{1 \le i \le L} k_i$. Let $\mathcal{G}$ be the associated family of embeddings (3), and $\mathcal{F}$ the associated family of distance functions, (2). Let $\{(x_i, x_i')\}_{i=1}^n$ be inputs, such that $\|x_i\|_2, \|x_i'\|_2 \le b$ for all $i \le n$. Set $\alpha_i = \|\mathcal{A}_i\|_{op}$, $\beta_i = \|\mathcal{A}_i\|_{2,1}$ and $\rho_i = \|\phi_i\|_{Lip}$ for $i \le L$. In addition, set $a = \|\mathcal{A}_L\|_{2,\infty}$. Then*

$$\mathfrak{R}_n(\mathcal{F}) \le O\left( \frac{8bk}{n}\left(\rho_L a \prod_{i=1}^{L-1} \rho_i \alpha_i\right) + \frac{8kb^2 \log(2W)\log n}{\sqrt{n}}\left(\rho_L a \prod_{i=1}^{L-1} \rho_i \alpha_i\right)^2 \left(\sum_{i=1}^{L-1} \frac{\beta_i^{\frac{2}{3}}}{\alpha_i^{\frac{2}{3}}} + 1\right)^{\frac{3}{2}} \right).$$

(20)

$$\mathfrak{R}_n(\mathcal{F}) \le O\left( \frac{8\|\phi_L\|_\infty k}{n} + \frac{8\|\phi_L\|_\infty kb \log(2W)\log n}{\sqrt{n}}\left(\rho_L a \prod_{i=1}^{L-1} \rho_i \alpha_i\right) \left(\sum_{i=1}^{L-1} \frac{\beta_i^{\frac{2}{3}}}{\alpha_i^{\frac{2}{3}}} + 1\right)^{\frac{3}{2}} \right).$$

(21)

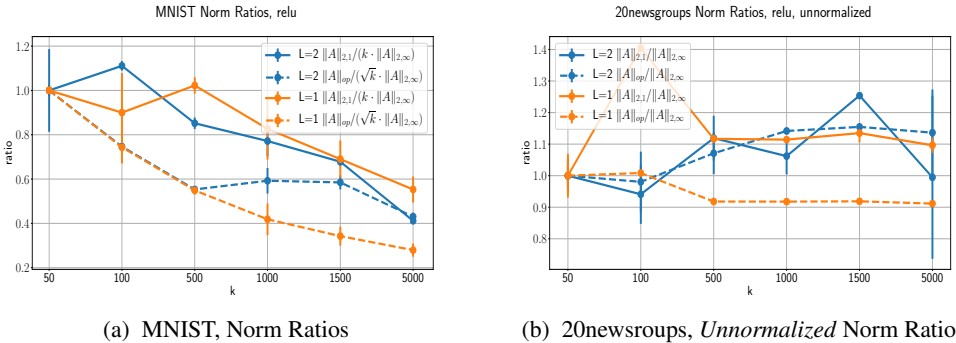

(a) MNIST, Norm Ratios

(b) 20newsroups, *Unnormalized* Norm Ratios

Figure 2: Dense and Sparse Regimes Demonstrated on MNIST and 20newsgroups Data

The approach we take here in Theroem 2 is as follows: Note that every element $f \in \mathcal{F}$ can be written as a sum

$$f_A(x, x') = \|g_A(x) - g_A(x')\|_2^2 = \sum_{j=1}^{k} f_{j,A}(x, x'), \tag{22}$$

where $f_j(x, x') = (g(x)_j - g(x')_j)^2 \in \mathbb{R}$ is the squared difference on the $j$-th coordinate. For a fixed $j \leq k$, denote by $\mathcal{G}'_j$ the family of the coordinate functions, $g_j : \mathbb{R}^d \to \mathbb{R}$, where $g \in \mathcal{G}$, and by $\mathcal{F}'_j$ the corresponding family of differences $f_j(x, x')$. Note that $\mathcal{G}'_j$ is induced by the same matrix tuples $\mathcal{A}$ as the full $\mathcal{G}$, with the exception of the last layer. Similarly to the arguments in Theorem 1, we can obtain bounds on $\mathfrak{R}_n\left(\mathcal{G}'_j\right)$ from the results in Bartlett et al. (2017). However, then we proceed differently. We first bound $\mathfrak{R}_n\left(\mathcal{F}'_j\right)$ in terms of $\mathfrak{R}_n\left(\mathcal{G}'_j\right)$ (Lemma 9, supplementary material), and then use the decomposition (22) to obtain that $\mathfrak{R}_n\left(\mathcal{F}\right) \leq \sum_{j=1}^{k} \mathfrak{R}_n\left(\mathcal{F}'_j\right)$. The full details are given in supplementary material Section C.

## 5 EXPERIMENTS

In this section we are interested in the following question: How do the matrices $A_L$ – the weight matrices of the last layer of the network – look for networks trained on standard datasets, with standard training procedures. In particular, do we obtain sparse or dense weights?

Recall that the notion of "dense" in this paper refers the specific condition that

$$\|A\|_{2,1} \geq const \cdot k \cdot \|A\|_{2,\infty} \text{ and } \|A\|_{op} \geq const' \cdot \sqrt{k} \cdot \|A\|_{2,\infty} \tag{23}$$

holds over a range of values $k$. That is, we look at several neural networks trained with the metric learning cost (12) on a fixed dataset, where all networks have the same architecture except the size of the output layer $k$. If the condition (23) holds[3] for a range of values $k$, we say that the problem is in the *dense regime*. For such a problem, the bounds of Theorem 2 would asymptotically be stronger than the bounds of Theorem 1. If the condition (23) fails, that is, for instance, the ratio $\|A\|_{2,1} / (k \cdot \|A\|_{2,\infty})$ exhibits strong decay with $k$, then we say that the problem is in the *sparse regime*, and the bounds of Theorem 1 would be better.

We demonstrate that for MNIST data, trained with SGD and no regularization, i.e. the standard deep learning training procedure, the weights are dense to a good approximation. On the other hand, for the 20newsgroup dataset, trained with $\ell_2$ regularization, the weights are sparse.

MNIST (mnist Dataset (2005)) is the standard dataset of handwritten digits and the 20newsgroups (20newsgoups Dataset (2010)) consists of newsgroups emails, labeled according to the group. To make computation and illustrations simpler, 20newsgroups is restricted to the first 10 labels.

To perform the optimization, we sample feature/label pairs $(x, l)$, $(x', l')$ independently from the train set, and minimize the loss using SGD on batches of such pairs, until convergence. The precise

---

[3]As discussed in Section 1.1, we have shown that the first condition in (23) implies the second, for large $k$. Nevertheless, as a sanity check, in what follows we evaluate both conditions.

loss that we use is

$$\ell(x, x', l, l') = 9 \cdot \rho(x, x') \cdot \mathbb{1}_{\{l=l'\}} + ReLU(1 - \rho(x, x')) \cdot \mathbb{1}_{\{l \neq l'\}}, \tag{24}$$

where $\rho(x, x')$ is the distance after the embedding, given by (1). This loss is a version of the loss $\ell_{S,D}(a, y)$ introduced in Section 2, with $S = 0, D = 1$, where the same class case is weighted by 9, to compensate for the fact that $\{l \neq l'\}$ pairs appear 9 time more frequently in the data. Additional details on the setting and the experiments, as well as the implementation source code, are given in supplementary material Section D.

We now describe the results in more detail. Consider first the single layer architecture, $L = 1$, with relu activation. That is, we train single layer networks of size $d \times k$, where $d$ is the problem's original feature dimension, and $k$ varies in the range 50 to 5000. For each trained network, we evaluate the quantities of interest – the ratios $\|A\|_{2,1} / (k \cdot \|A\|_{2,\infty})$ and $\|A\|_{op} / (\sqrt{k} \cdot \|A\|_{2,\infty})$ for MNIST, and unnormalized ratios $\|A\|_{2,1} / \|A\|_{2,\infty}$ and $\|A\|_{op} / \|A\|_{2,\infty}$ for 20newsgroups (see details below). These experiments are represented by orange lines in Figures 2a,2b.

Figure 2a represents the results for the MNIST dataset. In particular, the solid orange line shows the ratio $\|A\|_{2,1} / (k \cdot \|A\|_{2,\infty})$ and the punctuated orange line shows the ratio $\|A\|_{op} / (\sqrt{k} \cdot \|A\|_{2,\infty})$. To simplify the comparison, since we are only interested in change w.r.t $k$, all curves are normalized to have value 1 at $k = 50$. In such plot, the perfect dense regime behaviour would look as a straight horizontal line with value 1 for all $k$. While the actual lines exhibit some decay, this decay is slow. Compared to $k = 50$, at $k = 5000$ the orange lines drop by a factor of 2.5, while $k$ grows by a factor of 100. Thus we conclude that in this case the norms are well described by the dense regime.

Next, Figure 2b represents the results for the 20newsgroups dataset. While MNIST is generally nicely behaved, the 20newgroups is not. In particular, it consists of about $n = 7500$ samples in dimension $d = 15000 > n$, and unregularized SGD training would severely overfit the train data. Thus in this case we use the simplest possible regularization term, the $\ell_2$ regularization $\lambda \cdot \|A_1\|_{2,2} / k$, with $\lambda = 0.01$ in all experiments.

The orange lines in Figure 2b describe the ratios $\|A\|_{2,1} / \|A\|_{2,\infty}$ (solid) and $\|A\|_{op} / \|A\|_{2,\infty}$ (punctuated). Note that these ratios are *not* normalized by $k$. The curves are still normalized to be 1 at $k = 50$. Thus, the approximately straight lines we observe in Figure 2b indicate that the above norm ratios practically do not change as $k$ grows, implying that the problem is firmly in the sparse regime.

The blue lines in Figure 2 describe similar experiments for a two layer architecture, $L = 2$. In these experiments, the size of the first layer $A_1$ was fixed, $d \times 500$, with leaky $relu(0.2)$ activation. The second layer, $A_2$, was of size $500 \times k$, varying with $k$. The regualrization term for 2-newsgroups was $\lambda \left( \|A_1\|_{2,2} / 500 + \|A_2\|_{2,2} / k \right)$, again with $\lambda = 0.01$. The ratios in Figure 2 were this case computed for the second layer, $A_2$. One can see that the conclusions for $L = 2$ are similar: MNIST is in the dense regime, while 20newsgroups is sparse.

Each experiment was repeated 6 times, and every value in Figure 2 is a mean over 6 outcome values. Every value also has an error bar which indicates the magnitude of the standard deviation around the mean. However, in most cases, these bars are small compared to the magnitude of the mean are not visible in the figures.

## 6 CONCLUSIONS AND FUTURE WORK

In this paper we have obtained the first generalization guarantees for multilayer non-linear metric learning. We have introduced two types of bounds, which are appropriate for sparse and non-sparse regimes of the weights $A_L$, and we have empirically shown that both regimes may occur in common SGD training settings.

We conclude with an open question: When there are two different bounds on the same quantity, it is natural to ask whether there is a third bound, that interpolates both bounds (i.e. is as strong as both of them) and has a simple form. As discussed in Section 3, for the linear case there indeed is such a bound. It thus would be of interest to understand whether one can obtain a similar strengthening for the non linear multilayer case.

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
