# OpenReview forum: "Two Regimes of Generalization for Non-Linear Metric Learning"
_ICLR.cc/2022/Conference — ICLR 2022 Submitted_

### Official Review · Reviewer_w4wy · 2021-10-30

**Correctness:** 4
**Technical Novelty And Significance:** 3
**Empirical Novelty And Significance:** 2
**Recommendation:** 6
**Confidence:** 3

**Main Review:**

**Strength**

The authors show that for the metric learning setting the bounds from Bartlett, Foster, and Telgarsky can be improved by checking if the last layer of the network is dense.

The authors then show that this setting where the last layer is dense occurs in practice.

The authors show that their bound matches known bounds for the linear case.

**Weakness**

While the bound is an improvement, I do not think the improvement is significant. This is my main concern. See the discussion in the questions section.

I would have liked more discussion on how the bounds translate to generalization error bounds? Specifically, are they meaningful empirically?

**Questions**

1) I am not sure how the bounds are dimension free. The depth $L$ explicitly occurs in the bound, we have products and sums of $L$ terms. Second, as we increase the width, unless we decrease the magnitude of each entries, the matrix norms will increase as well. Hence both factors the width and depth play a factor in the bound.

2) How big do we expect the matrix norms to be? Because that determines that size of the bound.

At initialization, (if we use something like LeCun initialization), we expect each column to have norm 1 (due to the normalization) and so we expect $\\|A\\|\_{2,1} \sim k$. Then from your lemma we expect $\\|A\\|\_{op} \sim \sqrt{k}$.   Then if we have a "square" network (so $k_i$ are all equal), then

$$\sum_{i=1}^L \frac{||A_i||\_{2,1}^{2/3}}{||A_i||_{op}^{2/3}} \sim Lk^{1/3}$$.

So ignoring the log factors, assuming 1-Lipschitz activation maps, we get that the bound from theorem 1 looks like

$$ \frac{b^2}{\sqrt{n}}L^{3/2}k^{L+1/2}$$

So first, this very explicit depends on $k$ and $L$. Second, an improvement of $\sqrt{k}$ is an improvement, however, if $L$ is large I don't think it is significant.

Okay so this was at initialization, but maybe during training the norms change significantly. Do the authors know if this occurs in practice. My concern is if we are in the large width limit, then the results from Neural Tangent Kernel theory will tells us that training didn't change the norms.

3) I do not fully follow the discussion about the quadratic dependence. Specifically, are the authors claiming that the complexity must be lower bounded by $\Pi_i \|A_i\|_{op})^2$? If so then the improvement in the bound is significant.






**Summary Of The Paper:**

In this paper, the authors look at the Rademacher complexity of the family of Euclidean metrics learned on a data set via an $L$ layer network where the activations functions are Lipschitz. The idea behind the proof is to use the bounds for $\epsilon$ net for the embedding network from Barlett, Foster, and Telgarsky 2017. This net, then provides a net for space of metrics with only requiring a change in the constants. Using this net, the authors then use standard arguments to bound the Rademacher complexity.

Specifically for the metric learning setting they show that this bound can be improved.

**Summary Of The Review:**

Overall the paper presents a nice adaptation of the bounds from Bartlett, Foster, and Telgarsky for the problem of metric learning. I think understanding metric learning is an important problem in machine learning. However, I am not convinced that the improvement in the bounds is significant. However, my opinion on this can be changed.

---

> ### Author Response · Authors · 2021-11-18
> **Author Response for review w4wy**
>
> We thank the reviewer for the thoughtful comments.
>
> Before discussing questions 1-3 in detail, we would like to make a few general points about the significance of the results:
> 1) Our results extends the existing results for the linear case. Since in comparison to the linear case the methods are completely different, if the existing results are of significance, ours probably should be too.
> 2) The approach to Theorem 2 combines the covering number bounds (similarly to Bartlett et al 2017, but properly adapted) with additional new arguments, which allow us to obtain the different bound.
> 3) Perhaps most importantly, the current general philosophy of "why there is generalization" is that "effectively" there is a small number of parameters. This is implied by the use of sparsity promoting norms such as  A_{2,1} in the literature, as discussed in the paper.  On the other hand, here we have shown that even in the dense case, one can say something non-trivial about generalization (that is, something that is not implied by the $\|A\|_{2,1}$ bounds). We have also shown that the dense regime occurs naturally during  unregularized SGD training.
> 4) Note also that the use of the $\|A\|_{2,\infty}$ norm, a norm that is suitable for dense situations, is new in metric learning, and in generalization literature in general.
>
>
> We now address the questions in detail.
>
> >Question 1:  I am not sure how the bounds are dimension free...
>
> While very common, the term "dimension free" is indeed somewhat ambiguous. We use it in the standard sense -- that dimensions enter the bound only through norms.
> Then for fixed $n$, the bound can be small even in high dimensions. (in contrast to, say, standard VC-dim bounds). In that sense eq. (7) is dimension free, but (8) is not. As discussed in the paper, this does not necessarily mean that either bound is better than the other. Note that partly due to the above ambiguities, we do not use the "dimension free" terminology in this paper, and only mention it once in passing.
>
>
>
> >Question 2:
>
> Before discussing the typical magnitudes, it is worth noting that in principle, one can *force* the bounds to be as small as one likes. This can be done for instance by using appropriate regularization during training (but see also [*]).
>
>
> Returning to the calculation presented in the question: When $\|A\|_{op}$ is not regularized, it will indeed be of order $\sqrt{k}$, and
> the whole bound will be $\sim k^{L/2}$, i.e. exponential in $L$, and therefore unlikely to be useful for large $L$. To the best of our knowledge, at the moment, *all* existing bounds for DNNs are exponential in $L$.
>
> That said, our work concentrates on the characteristics of the problem that are specific to the metric learning setting. In that sense, the question of whether DNN bounds are exponential in $L$ is somewhat orthogonal to what we study here.
>
> In addition, note that metric learning embeddings do not necessarily have to be deep. Already linear embeddings, i.e. $L=1$ and $\phi(x)=x$ have a sizeable literature of  applications. In the final version of this paper we will include examples showing that by adding a non-linearity, one can improve the performance even in $L=1$ case, and improve it further with $L=2$.  See also [**].
>
> In such cases, $\sqrt{k}$ can be significant, and in general our theory indicates that, contrary to the standard theory, one does not necessarily have to regularize and force sparse solutions to obtain generalization.
>
> Does this address the concern of the reviewer for this question?
>
> >Question 3:
>
> In the paper we show that for the ReLU activation, the bound on the Rademacher complexity must be quadratically homogeneous in the weights $\|A_i\|$.  That is, if the bound on the weights, say $\|A_1\|$, grows by a factor of $u$, then the Rademacher complexity must grow by a factor of $u^2$. Such homogenity means that the *squares* in bounds (17), (20) are unavoidbale in the general case.
>
> We then show that for the bounded activations, the quadratic dependence can be avoided.
>
> That said, we believe it indeed can be shown that $\prod \|A_i\|_{op}^2$ is a lower bound, as suggested in the review. This can be done by a simple extension of the corresponding (non quadratic) lower bound in Bartlett et al 2017. That would be a somewhat stronger statement than what is in the paper at the moment. But again, our goal was only to show that square dependence is unavoidable, for which current arguments suffice.
>
> [*] Of course, if the weight values are too strongly constrained, then the performance on the *train* set would be poor. That is, the generalization gap would be small (generalization gap is the difference between train and test errors), but both train and test errors would be big.
>
> [**] Consider two concentric circles in R^2, with inner and outer circles belonging tio different classes. A linear embedding can not drive the cost (12) to zero, but a non-linear one can, with L=1 and high enough k.

---

> > ### Comment · Reviewer_w4wy · 2021-11-19
> > **Thanks for the clarifications.**
> >
> > So if I understand correct for a ReLU network we that the Rademacher complexity is $\Omega(\Pi |A_i|^2)$. In this case, the improvement by $\sqrt{k}$ is indeed significant.
> >
> > Similarly the authors are right, for small $L$ the improvement is significant as well!.
> >
> > I will increase my score.

---

> > > ### Author Response · Authors · 2021-11-19
> > > **Thanks!**
> > >
> > > Thank you for increasing the score!
> > >
> > > Indeed, $\Omega( \prod |A_i|_{op}^2 )$ is a lower bound for ReLU networks.
> > > We have now added a revision of the paper, where this is formally stated and proved.
> > > At the moment this material is in Section E in the Appendix.

---

### Official Review · Reviewer_Whaa · 2021-10-31

**Correctness:** 3
**Technical Novelty And Significance:** 3
**Empirical Novelty And Significance:** Not applicable
**Recommendation:** 3
**Confidence:** 3

**Main Review:**

**Caveats:**
- I am not familiar with the literature in this area, so cannot determine the novelty of the results or whether there are missing citations.
- I did not read the proofs in detail, so cannot vouch for the correctness of the mathematical results.


**Review:**

In general I found this paper quite difficult to read. This is in part because I am not an expert in the area of the paper, but also because it is somewhat confusingly organised.

To be explicit:
- after a brief introduction, the paper states in section 1.1 the bounds which are the main results, only partially defining the mathematical terms being used, and without context explaining why these are interesting.
- Then in section 1.2, there is a brief discussion of existing literature whose methods they build on.
- In Section 2, metric learning in general is once more discussed, and in section 3 other literature is again discussed.
- Much of the crucial mathematical notation used at the beginning of the paper is only defined in Section 4 where the results are formally stated.

I think the paper would benefit significantly from being reorganised. I suspect that this could lead to some space saving as there is currently some repetition.


On the technical side, it is not easy to pick apart the exact contributions of the paper. [I am not an expert in this area, so anyway wouldn't be fully able to evaluate the novelty of the results.] For instance:
- The results are all about proving bounds on Radermacher complexity. Why is this actually interesting? It is never stated explicitly in the paper.
- The bounds from Bartlett et al 2017 are mentioned multiple times. What exactly is the difference between those bounds and the ones in this paper?
- What is the connection between these results and the 'real world'? What are the limitations?


My review should be taken with a pinch of salt because I am not an expert in this area, but in my opinion the paper is not ready for publication in its current form.



**Other Detailed comments:**

Abstract:
- I wasn't familiar with (2,1) norms and (2, \infty) norms. If other reviewers also complain of this, could you consider adding a couple of words to give more context, maybe even just "(2,1) matrix norm" or something?


Page 1:
- Last paragraph: I found this sentence a bit confusing, consider rephrasing? "...difference between the expected value of the loss and the average value of the loss on the train set". Did you mean, "expected value of the loss wrt the unknown population distribution vs empirical average over the samples in the train set"?
- 20newsgroups: could you cite the dataset here on its first mentioning in the main text?


Page 2:
- After eq (2): Given that the main contribution of the paper is proving upper bounds on the Radermacher complexity of this function class, I think it wouldn't be a bad idea to talk in a little more detail about the importance of Radermacher complexity.
- Just after equation (3): I think can be helpful to readers to connect the assumptions made with real settings. Could you maybe add a note saying which types of network architectures this framework does and does not encompass? (Without thinking too hard, I guess it does include convolutions and MLPs without biases, but not ResNets, and does include ReLUs, sigmoids. Are there any commonly used activations that are not Lipschitz? Can biases somehow be included WLOG?)
- Before eq (4): could you define the spectral/operator norm?
- In eq (4): I found this equation quite confusing for a while and thought there was a mistake with the subscript "i"s. It would probably make it a lot clearer if you define \mathcal{A}_i (I guess this should be the set of all A_i) and then make it sup_{A_i \in \mathcal{A}_i}.
- eq (5): what does the "O with a line on top" mean? Could you please define it? (maybe this is standard in this part of the literature and I'm just unfamiliar with it)
- "Here b is an upper bound on inputs" I'm not sure what this means. Also, by 'inputs' do you mean 'features'? (I presume you mean the norm of the features, but please be explicit.)



Page 4:
- eq (12) as someone not so familiar with this area, I wondered why the margin constants are called S and D?

Page 6:
- Can the assumption that \phi(0)=0 be made WLOG? If not, what generality is lost?
- Equation 16: In general I found the organisation of this paper a bit confusing, and this is a good example. Why does this definition not appear somewhere near the beginning of the paper?

Page 7:
- RelU -> ReLU

Page 8:
- 'Recall that the notion of “dense” in this paper refers [to] the specific condition that' -> I think this is the first time that the definition is actually given in this way. I think it would be clearer to properly define it the first time you use it on page 3.
- Something has gone wrong with the formatting of citations for MNIST and 20newsgroups datasets. Could you move the citations to the first time you mention the datasets?


**Summary Of The Paper:**

The authors consider the setting of metric learning. They prove bounds on the Radermacher complexity of embeddings-composed-with-distance-functions for a particular architecture of neural network, with different results for the 'dense' and 'sparse' regimes.


**Summary Of The Review:**

Paper is not fit for publication in its current form: it should be structured more coherently and the results discussed in more context.

---

> ### Author Response · Authors · 2021-11-18
> **author response for review Whaa**
>
> We thank the reviewer for taking the time to provide detailed comments.
>
>
> >Much of the crucial mathematical notation used at the beginning of the paper is only defined in Section 4 where the results are formally stated.
>
> We understand the possible issue with a presentation form in which the results are first discussed without introducing the formal notation.  However, this is absolutely the standard form of presentation for theoretical papers. It is designed this way so that the experts can get the gist of the paper as quickly as possible. Thus in the introductory sections 1-3, we only define notions if they are new to this paper (such as the $\|A\|_{2,\infty}$ norm, for instance), or are not completely standard.
>
>
>
> >The results are all about proving bounds on Rademacher complexity. Why is this actually interesting? It is never stated explicitly in the paper.
>
> The Rademacher complexity is one of the most central notions in the study of generalization, please see, for instance, the book [1] (or start with the wikipedia entry, [2]).
>
> In short, Rademacher complexity can be used to bounds the difference between the test (or population) and train errors, i.e. to bound the generalization errors. In fact, we do mention this relation in the paragraph above section 1.1, and in Section 2, "Background". In particular, the last inequality of Section 2 (unnumbered) provides the standard connection between the generalization bounds and the Rademacher complexity.
>
>
> >The bounds from Bartlett et al 2017 are mentioned multiple times. What exactly is the difference between those bounds and the ones in this paper?
>
> The bounds in Bartlett et al 2017 are for classification problems. Our bounds are for the metric learning problems. These are somewhat different types of problems, they are defined on different spaces, and the metric learning costs have specific structure that the classification costs do not have. We analyze this structure differently in theorems 1 and 2.
> The relation between our bounds and those of Bartlett et al 2017,  is that in our proof of Theorem 1, *a part* of the argument is similar to the argument in Bartlett et al. 2017. As a result, an additional similarity between the results is that both use "sparsity based norms", $\|A\|_{2,1}$.
>
>  In Theorem 2 we show that there is also a different bound, using the $\|A\|_{2,\infty}$ norm, that is better for dense weight matrices.
>
> Bounds for metric metric learning were previously available only for the linear case,
> and with different methods. Thus both our theorems extend the results to non-linear cases, and provide approaches to the metric learning problem that were not considered in the literature before.
>
>
> We will take the additional comments provided in the review into account when preparing the final version of the paper.
>
>
> [1] Mohri et al, Foundations of machine learning. 2018.
>
>
> [2] https://en.wikipedia.org/wiki/Rademacher_complexity

---

> > ### Comment · Reviewer_Whaa · 2021-11-22
> > **Thank you for the response**
> >
> > Thanks for the response. I hope that the authors will indeed take the feedback into account. For me personally the issues are too large that I would be comfortable increasing my score and recommending acceptance; I would encourage the reviewers to resubmit and updated manuscript.

---

> > > ### Author Response · Authors · 2021-11-22
> > > **What are the issues that were not addressed in the response?**
> > >
> > > We thank the reviewer for getting back to us. However, it is not clear what is meant by
> > > >For me personally the issues are too large
> > >
> > > What issues were not cleared by the above response? Are there specific problems with the paper that we can address?
> > >
> > > As mentioned earlier, we will take the feedback into account, and we thank the reviewer again for providing it. At the same time, it must be noted the original review's feedback is overwhelmingly concerned with the form of the presentation, and with clarification requests about topics that are commonly considered standard in the field. As we have discussed in the first response, our general form of presentation is standard, and we have explained the commonly accepted rationale behind it. Thus, while we will definitely fix the typos and rephrase several sentences pointed out in the review, we believe changing the order of the presentation would go
> > > against expectations of most readers.

---

### Official Review · Reviewer_nCbx · 2021-11-02

**Correctness:** 2
**Technical Novelty And Significance:** 1
**Empirical Novelty And Significance:** 2
**Recommendation:** 3
**Confidence:** 4

**Details Of Ethics Concerns:**

NAN

**Main Review:**

Strengths:
It is indeed a promising research direction to regard neural networks as a special nonlinear metric embedding.

Weakness:
1.	This paper only uses metric embedding to tell a story for DNN models and does not provide the specific relationship between metric learning and DNNs. For example, whether the feature transformation obtained by DNN meets the definition of metric (or part of the definition), and whether the perspective of metric embedding can bring new inspiration to the theory of DNNs.
2.	The metric learning theory in this paper basically comes from the generalization theory of neural networks [Bartlett et al. (2017)]. Compared with the previous theoretical results, the metric perspective analysis proposed in this paper does not give better results. From the existing content of this paper, the part of metric learning does not seem to work.


**Summary Of The Paper:**

This paper tries to provide uniform guarantees for the DNN type metric embeddings.

**Summary Of The Review:**

In a word, I think this paper does not provide a new theoretical guarantee for nonlinear metric learning, and its results are basically the same as the existing generalization theory of DNNs.

---

> ### Author Response · Authors · 2021-11-18
> **author response for review nCbx**
>
> >For example, whether the feature transformation obtained by DNN meets the definition of metric (or part of the definition),
>
> This is elementary. *Every* transformation composed with a Euclidean metric is a pseudo-metric. Depending on the cost, it may very well be of interest to have a pseudo metric embedding rather than the purely metric one. This would happen in cases where the loss does not distinguish between different points. They can then be identified as a single or nearly single point by the embedding.
>
> These considerations are standard. We refer the reviewer to the general literature on metric learning as discussed in Section 3.
>
>
> >This paper only uses metric embedding to tell a story for DNN models and does not provide the specific relationship between metric learning and DNNs.
>
> >whether the perspective of metric embedding can bring new inspiration to the theory of DNNs.
>
> We are unsure how to react to these comment. We provide a comprehensive mathematical analysis for an important problem.
>
>
>
>
> >The metric learning theory in this paper basically comes from the generalization theory of neural networks [Bartlett et al. (2017)].
>
> In this paper have proved two theorems about generalizaton bounds of metric learning costs with DNNs. Theorem 1 uses the sparsity based norms, and adapts existing covering number methods to the metric learning situation. Theorem 2 introduces a new, non-sparsity based norm, and uses a different, new argument to bounds the Rademacher complexity.
>
> Both theorems are new, and extend the existing results in metric learning. The approaches to the proofs, both of Theorem 1  and Theorem 2, have not been used before in the context of metric learning.
>
> If the reviewer believes any of the above statements to be inaccurate, please provide substantiated arguments, with references.
>
>
> >Compared with the previous theoretical results, the metric perspective analysis proposed in this paper does not give better results.
>
> We do not follow what the reviewer is asking us. Which previous theoretical results are meant here? Please provide references so we can address this. As discussed above, our results are new, and are the first results for the non-linear case.
>
>
> >From the existing content of this paper, the part of metric learning does not seem to work.
>
> It is not clear what is meant by "the part of metric learning does not seem to work". We think it works very well!
>
>
> > In a word, I think this paper does not provide a new theoretical guarantee for nonlinear metric learning, and its results are basically the same as the existing generalization theory of DNNs.
>
> We respectfully disagree. As discussed above, we proved two new theorems.
> The approach of Theorem 2 is completely new, while the argument of Theorem 1 is a significant modification and adaptation  of existing arguments. Further, we have introduced non-trivial results for the $\|A\|_{2,\infty}$ norm, a situation which has not been considered in the literature before.

---

### Official Review · Reviewer_zFNY · 2021-11-03

**Correctness:** 4
**Technical Novelty And Significance:** 3
**Empirical Novelty And Significance:** 2
**Recommendation:** 5
**Confidence:** 3

**Main Review:**

Strengths:

The paper exploits the structure of metric learning to extend the uniform generalization bound of Bartlett et al 2017 and a new 2-infinity bound for the metric learning setting.

Experiments show both bounds are useful depending on data/training.


Weaknesses:

Theorem 1 seems like straightforward application of Bartlett et al 2017

Novel technical insights for the new results e.g. Theorem 2 could be more clearly stated. E.g. the novel technical insight for theorem 2 is very briefly stated (e,g. it would be nice to have informal or formal statement of Lemma 9 in main paper).

The sparse and non-sparse regimes are defined based on when one bound dominates the other, but it is less clear what data or training properties can result in one or the other. E.g. what regime is expected when MNIST is trained with regularization or 20newsgroups without (or with less regularization). More thorough experiments say with more datasets and degrees of regularization can potentially give interesting insights.

**Summary Of The Paper:**

The paper provides two new generalization bounds for non-linear metric learning with deep neural networks, by extending results of Bartlett et al. 2017 to the metric learning setting. The two bounds have been called the 'sparse' and 'non-sparse' bounds and differ in the norm used for the last layer. Experiments are performed where it is shown that either bound may dominate on real datasets.

**Summary Of The Review:**

Overall the paper gives interesting generalization bounds for the metric learning setting but does not seem ready for publication.

---

> ### Author Response · Authors · 2021-11-18
> **author response for review zFNY**
>
> We thank the reviewer for the feedback. We believe however that
> they miss some points, and would appreciate feedback on our comments too.
>
> >Novel technical insights ... could be more clearly stated.
>
> Theorem 2 is indeed the main technical contribution of this paper. In the final version of the paper we will add Lemma 9 to the main text.
>
>
> >The sparse and non-sparse regimes are defined based on....
>
>
> We believe that an extended empirical investigation of the distribution of the weights in terms of density would indeed be of interest. In fact, we are somewhat surprised that we seem to be the first to consider these properties quantitatively in the experiments. However, similarly to a significant body of literature on metric learning in the linear case, this paper is mainly concerned with the theoretical bounds and extended experiments would be out of both scope and space constraints.
> With that in mind, our expectation is that without regularization, all the networks should be in the dense regime. This happens on both MNIST and  20newsgroups [*]. There seems to be no reason that sparsity should occur spontaneously [**], especially since the initialization is dense. This in particular motivated our interest in bounds that are not sparsity based like the (2,1) or (2,2) norms common in the literature. Our results are indeed the first in the generalization literature to use the $(2,\infty)$ norm.
>
> On the other hand, note that with regularization, we can achieve any desired level of sparsity, simply forcing the value of the norms via the penalty term. If the restrictions on the norms are too strong, we would obtain poor performance on the *train* set. Thus generalization would be good (i.e., small difference between train and test) but irrelevant. At the same time, on some datasets, training without regularization would lead to overfitting (i.e., poor test performance and dense matrices with extremely large norms). This happens on the 20newsgroups dataset, due to the large input dimension, as discussed in Section 5. Due to this, we train 20newsgroups with regularization.
>
>
> > Theorem 1 seems like straightforward application of Bartlett et al 2017
>
> We believe it would be quite difficult to decide objectively whether an argument indeed "seems like straightforward application of ..." or not.  If the reviewer believes this strongly, could you provide additional details as to why this is the case?
>
> However, in this regard we would like to mention the following points:
>
> 1) Building on some of the results of Bartlett at al, which build in turn on results of Zhang 2002 and Bartlett 1998, our proof of Theorem 1 is still 2 and half pages long, including the introduction of notions required specifically for the proof. And this with a fairly condensed writing.
>
> 2) The existing approaches to the *metric learning* problem are very different from the approach we take either in Theorem 1 or Theorem 2. As discussed in Sections 1.2 and 3, existing approaches to metric learning can not be extended even to the case $L=1$, with a *non-linear* activation. Thus in the context of metric learning, the approach is completely new.
>
> 3) The difficulty that the arguments in Bartlett et al 2017 aim to address is depth, i.e., how one bounds the behaviour complexity with increasing L. On the other hand, the problem that our paper addresses is how one treats the specific metric learning cost, with non-linear activations. This problem is completely orthogonal to the issues treated in Bartlett et al. All the essential features of both Theorem 1 and Theorem 2 can be stated and proved already in $L=1$ case with non-linear activations, with direct covering and other estimates (that is, without using Bartlett et al results, but using arguments developed here). Both results are arguably non-trivial already in this case. In view of this, viewing our results as "applications of Bartlett et al 2017" somewhat misses the point of what we do.
>
> 4) In view of the above, note that low depth metric embeddings can in fact be practically useful. Already linear embeddings have a sizeable literature of applications. In the final version of this paper we will provide simple examples showing that by adding a non-linearity, one can improve the performance even in $L=1$ case, and improve it further with $L=2$. See also [***]
>
> [*] If 20newsgroups is trained without regularization.
>
> [**] Perhaps sparsity could appear spontaneously in some special situations, such as long training, with sigmoid non-linearities, which could drive some of the norms up, while some neurons would remain "unused". But such special circumstances would be outside the scope of the paper.
>
> [***] Note that synthetic examples where non-linearity helps are easy to construct. Consider two concentric circles in R^2, with inner and outer circles belonging tio different classes. A linear embedding can not drive the cost (12) to zero, but a non-linear one can, with L=1 and high enough k.

---

### Decision · Program_Chairs · 2022-01-20

**Decision:**

Reject

**Comment:**

The paper provides two new generalization bounds for non-linear metric learning with deep neural networks, by extending results of Bartlett et al. 2017 to the metric learning setting. The main contribution of the paper is by extending the techniques of Bartlett et al. from a classification setting to the metric learning setting (which has very different objectives) and consider two regimes. In the first regime the techniques are fairly similar but the second regime is more novel. However, the current version of the paper does not highlight the similarity and differences between the results and techniques with Bartlett et al. 2017; it also does not give sufficient intuition on how the metric learning setting is fundamentally different from the classification setting and how the paper leverage the difference to get improved bounds. All the reviewers had some confusions to different degrees, and the paper would be much stronger if it can explain the intuition and make more explicit comparisons.